# POWERSOFTMAX: TOWARDS SECURE LLM INFERENCE OVER ENCRYPTED DATA

## ABSTRACT

Modern cryptographic methods for implementing privacy-preserving LLMs such as Homomorphic Encryption (HE) require the LLMs to have a polynomial form. Forming such a representation is challenging because Transformers include non-polynomial components, such as $\mathrm{Softmax}$ and layer normalization. Previous approaches have either directly approximated pre-trained models with large-degree polynomials, which are less efficient over HE, or replaced non-polynomial components with easier-to-approximate primitives before training, e.g., $\mathrm{Softmax}$ with pointwise attention. The latter approach might introduce scalability challenges.

We present a new HE-friendly variant of self-attention that offers a stable form for training and is easy to approximate with polynomials for secure inference. Our work introduces the first polynomial LLMs with 32 layers and over a billion parameters, exceeding the size of previous models by more than tenfold. The resulting models demonstrate reasoning and in-context learning (ICL) capabilities comparable to standard transformers of the same size, representing a breakthrough in the field. Finally, we provide a detailed latency breakdown for each computation over encrypted data, paving the way for further optimization, and explore the differences in inductive bias between transformers relying on our HE-friendly variant and standard transformers. Our code is attached as a supplement.

## 1 INTRODUCTION

Privacy-Preserving Machine Learning (PPML) solutions and in particular privacy-preserving LLMs Yan et al. (2024); Yao et al. (2024) aim to provide confidentiality guarantees for user data, the model owner, or both. One prominent cryptographic primitive for achieving this is HE, as it allows computations to be performed on encrypted data without revealing any information to the (potentially untrusted) computing environment. Furthermore, it enables non-interactive computations, which increases the usability of these solutions.

However, modern HE schemes like CKKS Cheon et al. (2017) face a significant challenge of only supporting polynomial computations on encrypted data. This limitation complicates the deployment of DL models in HE environments, particularly for LLMs, which depend on non-polynomial functions like $\mathrm{Softmax}$ in self-attention. To overcome this, existing approaches have adapted these non-polynomial operations into polynomial forms using techniques such as unique polynomial approximation Lee et al. (2021) or fine-tuning procedures Baruch et al. (2022). While these methods have enabled the execution of FFNs, CNNs Baruch et al. (2023); Lee et al. (2022), and small transformers Zimerman et al. (2024); Zhang et al. (2024b) over HE, they often struggle with stability and sensitivity issues Zhou et al. (2019); Goyal et al. (2020), preventing an effective scale-up.

We take a different approach. Rather than modifying existing transformers to fit within the constraints of HE, we revisit the core design principles of the transformer architecture Vaswani et al. (2017) through the lens of the CKKS constraints. Concretely, we ask:

*Are there HE-friendly operators that can replicate the key design principles of self-attention?*

We find a positive answer by introducing a power-based variant of self-attention that is more amenable to polynomial representation. Models with this variant maintains comparable performance to $\mathrm{Softmax}$-based Transformers across several benchmarks and preserve the core design characteristics of self-attention. We also present variants that include length-agnostic approximations or

improved numerical stability. The entire mechanism offers a more HE-friendly and effective transformer solution than previous approaches, enabling our method to scale efficiently to LLMs with 32 layers and 1.4 billion parameters.

**Our main contributions:** (i) We propose a HE-friendly self-attention variant tailored specifically for HE environments. This variant minimizes the usage of non-polynomial operations while maintaining the core principles of attention mechanisms. Additionally, we extend this approach by introducing a numerically stable training method and a length-agnostic computation strategy for inference. As a result, our model enables secure inference at scale and is more efficient than existing methods. (ii) We leverage this technique to develop a polynomial variant of RoBERTa and the first polynomial LLM that exhibits reasoning and ICL capabilities, as well as the largest polynomial model trained to date, encompassing 32 transformer layers and approximately a billion parameters. (iii) We provide early ablation studies and profiling of latency breakdowns over encrypted data, paving the way for further improvements.

## 2 BACKGROUND

**Homomorphic Encryption (HE).** A form of encryption that enables processing of encrypted data without decrypting it Gentry (2009), so that after decryption the results are similar to the results of applying the same computation on the unencrypted inputs. Some HE schemes Brakerski et al. (2014); Fan & Vercauteren (2012) are exact, meaning that the value of the decrypted ciphertext is exactly the result of the arithmetic operation, while some like CKKS Cheon et al. (2017) are approximate and introduce a tiny amount of noise ($\epsilon$) to the decrypted values. Formally, an HE scheme encryption operation $E : \mathbb{R}_1 \to \mathbb{R}_2$ takes a plaintext from a ring $\mathbb{R}_1(+, *)$ and transforms it into a ciphertext in a ring $\mathbb{R}_2(\oplus, \odot)$ (and the opposite holds for decryption $D : \mathbb{R}_2 \to \mathbb{R}_1$). All while also maintaining the following properties for an input $x, y \in \mathbb{R}_1$: (i) $D(E(x)) = x + \epsilon$, (ii) $D(E(x) \oplus E(y)) = x + y + \epsilon$, and (iii) $D(E(x) \odot E(y)) = x * y + \epsilon$.

**Polynomial Deep Learning Models.** Deep learning models rely heavily on non-polynomial activation functions like $\mathrm{ReLU}$, sigmoid, and tanh to introduce non-linearity, which enhances model expressiveness. However, over most HE schemes, operations must have a polynomial form. Prior work has reported that polynomial DNNs tend to face instability as the network grows (Zhou et al. (2019); Goyal et al. (2020); Chrysos et al. (2020); Gottemukkula (2020)). Thus, maintaining an accurate and stable network when using polynomial approximations is challenging.

There are two primary approaches for polynomial approximation: Post-Training Approximation (PTA), and Approximation-Aware Training (AAT). In PTA, the approximation is applied to a pretrained network without modifying the model architecture and parameters (Lee et al. (2021); Ao & Boddeti (2024); Ju et al. (2023); Zhang et al. (2024b)). This approach saves the costly training process by providing a precise approximation for each computation using high-degree polynomials.

In contrast, AAT aims to reduce the number of required approximation polynomials in the network or to minimize their degree Gilad-Bachrach et al. (2016); Lee et al. (2023); Baruch et al. (2022; 2023); Ao & Boddeti (2024); Drucker & Zimerman (2023); Zimerman et al. (2024). Doing so can improve both latency and precision under HE, as higher-degree polynomials increase the *multiplicative depth*–the number of sequential multiplications required–leading to higher computational overhead, greater resource consumption, and increase the accumulated noise. Typically, this is achieved by modifying the network architecture. For instance, early studies in this area substituted the $\mathrm{ReLU}$ activation function with quadratic activations Gilad-Bachrach et al. (2016); Baruch et al. (2022).

To reduce polynomials' degrees in large-scale models, such as ResNet152 on ImageNet and transformers, while still achieving accurate approximation, recent works (Baruch et al. (2023) and Zimerman et al. (2024)) have suggested using the training process to minimize the input range to the non-polynomial operations. This is done by adding a **range-loss term** to the original loss function, encouraging the network to operate within a range where lower-degree polynomial approximations are accurate enough.

**Polynomial Transformers.** To enjoy the non-interactive property of HE-based solution, this paper only considers fully polynomial models. While other secure alternatives such as Chen et al. (2022); Ding et al. (2023); Liu & Liu (2023); Liang et al. (2024); Gupta et al. (2023); Zheng et al. (2023) exist, they require interaction with the user to process non-polynomial operations. This involves extra

communication overhead and may also be susceptible to some cryptographic attacks Akavia & Vald (2021). In contrast, the use of HE enables non-interactive computation in untrusted environments without additional communication. In transformer architectures, the $\mathrm{Softmax}$ function (which involves exponentials and divisions), $\mathrm{LayerNorm}$, and GELU are non-polynomial operations that need to be replaced or approximated.

The first work to present a fully polynomial transformer was by Zimerman et al. (2024), who used the AAT approach and substituted $\mathrm{Softmax}$ with a $\mathrm{scaled\text{-}ReLU}$ that is easier to approximate by polynomials. They also used the range-loss term during training to reduce the polynomial degree required for accurate approximation of $\mathrm{ReLU}$ and $\mathrm{LayerNorm}$. They demonstrated a 100M-parameter polynomial transformer pretrained on WikiText-103 for secure classification tasks using HE.

Alternatively, Zhang et al. (2024b) used the PTA approach. They introduced a polynomial transformer by directly approximating the numerator, denominator, and division separately, without dedicated training modifications. However, as described in Sec. 5.3, this approach has disadvantages in terms of latency and scalability.

In this work, we scale up the AAT for transformers approach, by replacing $\mathrm{Softmax}$ with a polynomial-friendly alternative, that closely replicates its behavior. This enhancement allows us to improve model performance and scalability, enabling the deployment of $1.4$B-parameters LLMs under HE, while maintaining the model's performance. After training, we approximate the non-polynomial operations using methods detailed in Appendix D, converting the trained model into a polynomial form for secure inference.

## 3 PROBLEM SETTINGS

This work focuses on secure inference for LLMs over HE, aiming to obtain a polynomial representation in the final model rather than addressing secure training procedures. Specifically, we target scenarios where either the model's weights or the input samples are encrypted during inference. Achieving this goal requires developing a transformer variant that relies exclusively on polynomial computations while matching the language modeling capabilities of transformers with billions of parameters trained on a trillion tokens. This problem is particularly challenging because polynomial networks tend to face instability issues from both theoretical and empirical perspectives Zhou et al. (2019); Goyal et al. (2020); Zhang et al. (2024a), even at scales much smaller than those considered in this work. Moreover, as the degree of the polynomials increases, both the accumulated noise and computation time during secure inference rise significantly, often yielding impractical solutions. Therefore, a key challenge lies in minimizing the degree of each polynomial layer and reducing the model's overall multiplicative depth.

## 4 METHOD

The self-attention mechanism in transformers is defined by:

$$\text{Self-Attention}(Q, K, V) = \text{Softmax}\left(\frac{QK^T}{\sqrt{d_k}}\right) V \tag{1}$$

which is inherently non-polynomial because it includes division and exponential operations. Furthermore, for numerical stability it is common to compute the $\mathrm{Softmax}$ function using the *log-sum-exp* trick, which adds non-polynomial operations. For example, it involves calculating the maximum absolute values of each row of $QK^T$. The latter operation involves high-degree polynomials that in HE environments may introduce significant noise. Instead of directly approximating the maximum, division, and exponential functions individually (as done in Nexus Zhang et al. (2024b)), our objective is to develop a more polynomial-friendly and HE-compatible $\mathrm{Softmax}$ variant for transformers. Such a mechanism can not only reduce the overall computational complexity, particularly in terms of multiplication depth, but also supports scaling of polynomial transformers to models with billions of parameters and deeper architectures.

### 4.1 HE-FRIENDLY ATTENTION

To design a HE-friendly variant of $\mathrm{Softmax}$-based attention, we start by distilling its properties that correlate with its performance: (i) normalization of the attention scores ensures they are bounded in

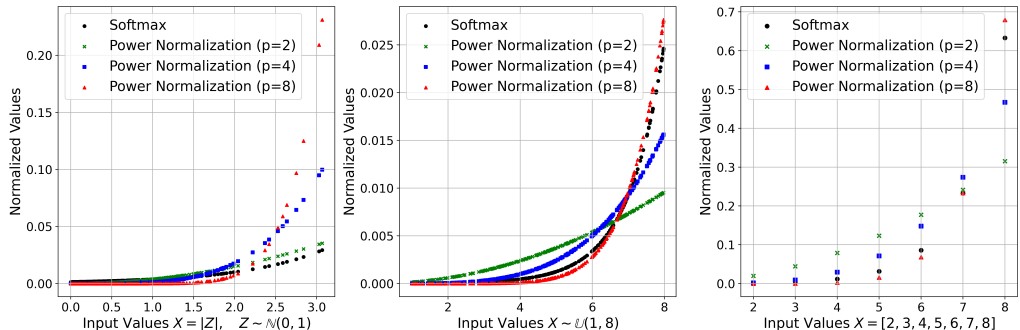

Figure 1: **Comparison of** Softmax **and** PowerSoftmax **normalization** on normally distributed values on the left, uniformly distributed values in the middle, and evenly spaced values on the right. As can be seen, the empirical scaling trends are relatively similar.

$[0, 1]$, with their sum equal to 1, similar to probabilities (ii) exponential scaling of attention scores, such that it amplifies the differences between higher and lower scores, and (iii) monotonic increasing and order-preserving behavior, meaning that higher input values yield higher output values while preserving the relative order of the input values. Building on these properties, we introduce the following attention variant:

$$\text{HE-Friendly Attn}(Q, K, V) = \text{PowerSoftmax}\left(\frac{QK^T}{\sqrt{d_k}}\right)V, \quad \text{PowerSoftmax}(x)_j = \frac{x_j{}^p}{\sum_i x_i^p} \quad (2)$$

where we replaced the $\text{Softmax}(x)_j = e^{x_j} / \sum_i e^{x_i}$ function with PowerSoftmax, for some positive even $p$. Eq. 2 describes a variant that satisfies #i, but not accurately retain properties #ii and #iii, as our variant performs *polynomial scaling* instead of *exponential scaling* (both have superlinear trends), and because it is not strictly monotonic increasing. Nevertheless, for suitable values of $p$, the polynomial scaling can mimic the trends of exponential scaling relatively well, as shown in Fig. 1. Additionally, instead of maintaining the order and strictly increasing monotonic, our variant preserves *the order of the norms* and is increasing monotonically for positive values.

To highlight the similarities and differences between both attention mechanisms in Eqs. 1 and 2, we introduce a generalization of the Softmax function within transformers, using an elementwise activation function $\sigma : \mathbb{R} \to \mathbb{R}$ followed by proportional normalization $\mathbb{N} : \mathbb{R}^L \to \mathbb{R}^L$:

$$\text{Generalized Self-Attn}(Q, K, V) = \mathbb{N}\left(\sigma\left(\frac{QK^T}{\sqrt{d_k}}\right)\right)V, \quad \mathbb{N}(\mathbf{x})_j = \frac{|\mathbf{x}_j|}{\|\mathbf{x}\|_1} \quad (3)$$

In this formulation, Softmax is obtained by setting $\sigma$ as $\sigma_e(x) = \exp(x)$, while our variant is defined by using $\sigma_p(x) = x^p$ for $\sigma$ using a positive even $p$.

### 4.2 $\frac{1}{\epsilon^2}$-Lipschitz Division for Softmax Approximation

A key challenge in approximating Softmax or Eq. 2 with polynomials is the behavior of the inverse term $1/x$, which grows rapidly near zero, i.e., $\lim_{x \to 0^+} \frac{1}{x} = \infty$. While Softmax deals with summation over strictly positive exponents, this property does not hold for PowerSoftmax, where the denominator can potentially reach zero. To address this, we propose the $\frac{1}{\epsilon^2}$-*Lipschitz division for Softmax*, modifying the denominator of $\mathbb{N}$ before training as:

$$\frac{1}{\epsilon^2}\text{-Lipschitz HE-Friendly Attn}(Q, K, V) = \mathbb{N}_\epsilon\left(\sigma_p\left(\frac{QK^T}{\sqrt{d_k}}\right)\right)V, \quad \mathbb{N}_\epsilon(\mathbf{x})_j = \frac{|\mathbf{x}_j|}{\epsilon + \|\mathbf{x}\|_1} \quad (4)$$

Here, $\epsilon$ (e.g., $1e-3$) ensures the denominator is bounded away from zero, preventing discontinuities and ensuring $\lim_{x \to 0^+} \frac{1}{x+\epsilon} = \frac{1}{\epsilon}$. This introduces a single non-polynomial division, which is $\frac{1}{\epsilon^2}$-Lipschitz continuity function, making the polynomial approximation more tractable. Importantly, unlike the common use of $\epsilon$ for numerical stability in division, our approach focuses on much larger values of $\epsilon$ to reduce the multiplication depth required for approximation, making the approximation problem significantly easier for secure inference over HE.

### 4.3 Stable Variant For Training

By examine the $i$-th row of the unnormalized attention scores $S_i = \left[ \frac{1}{\sqrt{d_k}} QK^T \right]_i$, it is clear that Eq. 2 and Eq. 6 can lead to training instability when applying PowerSoftmax, as when $|S_{i,j}| > 1$, $|S_{i,j}|^p$ can become very large, causing overflow, and when $|S_{i,j}| < 1$, $|S_{i,j}|^p$ can become very small, leading to underflow. In Transformers, a similar problem occurs with the traditional Softmax, which is mitigated using the *log-sum-exp trick* to scale the values of $|S_i|$ within a manageable range. Inspired by this, we propose a more stable version of our PowerSoftmax variant:

$$\text{Stable PowerSoftmax}(\mathbf{x})_j := \text{PowerSoftmax}\left(\frac{\mathbf{x}}{c}\right)_j, \quad c = \|\mathbf{x}\|_\infty + \epsilon' \tag{5}$$

This method leverages the fact that PowerSoftmax is invariant to division of its input by a constant $c > 0$ (similar to Softmax which is invariant under the subtraction of a constant). By selecting $c$ such that $\forall j |S_{i,j}| < 1$, we (i) ensure that the input values stay within a range where floating-point precision is more reliable ($0 < |S_{i,j}| < 1$), and (ii) stretch (or shrink) the values of $x$ to have a similar scale across different coordinates, preventing the loss of significant digits during division. Fig. 2 (middle) illustrates our HE-friendly training variant, built on top of Eqs. 4 and 5, compared to the original attention.

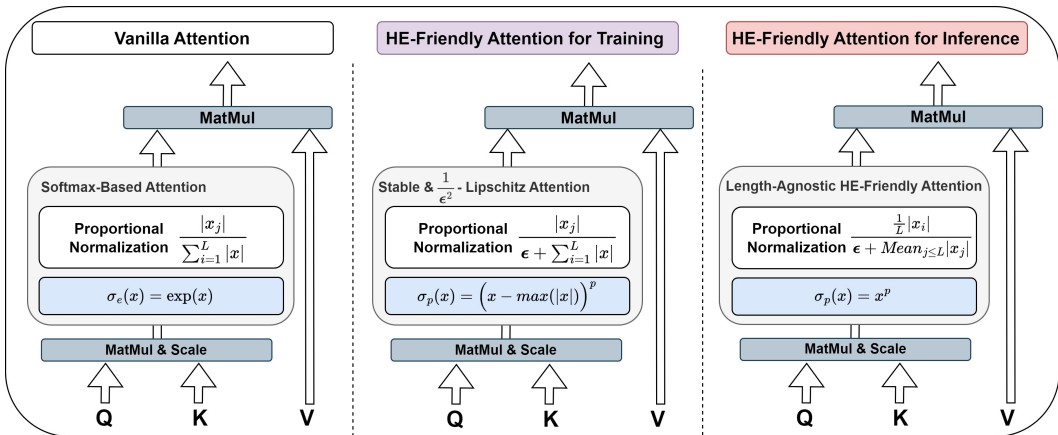

Figure 2: **Our Attention Variants:** (Left) the Softmax-based attention mechanism using the generalized attention formulation (Eq. 3). (Middle) Our variant for training (purple), builds on the stable variant from Eq. 5 and the Lipschitz division from Eq. 4. (Right) During secure inference with the polynomial model (red), we use a length-agnostic approximation for division, as described in Eq. 6.

### 4.4 Length-Agnostic Range for Polynomial Evaluation of Division

The only non-polynomial operation in Eq. 2 is division, which can be approximated effectively in a bounded domain using the Goldschmidt algorithm Goldschmidt (1964). However, in our attention variant, we need to approximate the function $\frac{1}{x}$, where x is the sum of the scores raised to the power of $p$, which is unbounded and increases linearly with the sequence length $L$. Thus, applying the Goldschmidt algorithm naively would struggle to precisely approximate division for both short and long sentences and would require relatively high-degree approximations due to the extremely large domain range. To address this problem, we propose a length-agnostic HE-friendly attention variant:

$$\text{Length-Agnostic PowerSoftmax}(\mathbf{x})_j = \frac{\frac{1}{L} x_j^p}{\text{Mean}_{i \leq L} x_i^p} = \frac{\left(\frac{x_j}{L'}\right)^p}{\text{Mean}_{i \leq L} x_i^p}, \quad L' = L^{\frac{1}{p}} \tag{6}$$

This variant leverages the fact that the sequence length $L$ is not a secret, and $\frac{1}{L}$ can be directly computed without approximation (or can be pre-computed by the client). This obtained approximation of division operates over the mean of the attention scores rather than their sum. Notably, assuming that the attention scores have a mean $\mu$ and variance $\sigma^2$, the asymptotic trends of these two approaches

when $L$ is increased can be described as follows (according to the law of large numbers):

$$\text{Mean } \sigma_p \left( \frac{1}{\sqrt{d_k}} Q K^T \right) \to \mu, \quad \sum \sigma_p \left( \frac{1}{\sqrt{d_k}} Q K^T \right) \to \infty \tag{7}$$

This reflects that our length-agnostic variant does not become more difficult to approximate as L increases, allowing us to present a more flexible and precise polynomial approximation. Fig. 2 (right) compares this variant with the original attention.

### 4.5 A Recipe for Polynomial LLM

Algorithm 1 illustrates the entire process, which is divided into three key stages: **(i) Architectural Modification:** We begin by modifying the original transformer architecture to use an HE-friendly attention variant (Eq. 5). This modified model is then trained from scratch using the same hyper-parameters as the vanilla transformer. **(ii) Range Minimization:** In the second stage, we apply a supplementary training procedure followed by Baruch et al. (2023) to ensure that the model operates within HE-friendly constraints. Specifically, we adjust the model's weights so that each non-polynomial component operates only within specific, restricted input domains. This is achieved by adding a regularization loss function that minimizes the range of inputs to non-polynomial layers. For activations and LayerNorm layers, we directly apply the method from Zimerman et al. (2024).

Additionally, for the HE-friendly attention mechanism, we introduce a tailored loss term defined as:

$$\mathbb{L}_{\text{PowerSoftmax}} := \sum_{n=1}^{N_L} \max_{c \in C} \left\{ |z|_{n,c}^i \right\} \tag{8}$$

where we denote the number of attention layers by $N_L$, the set of heads by $C$. Additionally, we denote the input at layer $n$ to the PowerSoftmax layer, at head $c \in C$, when the model processes the $x_i$ example by $z_{n,c}^i$. This loss serves two main purposes: First, it minimizes the upper bound of the denominator in the HE-friendly attention variant, making the approximation problem more tractable. Second, we observed that when the input norm to the HE-friendly attention is not too high, the stabilize factor defined in Eq. 5 can be omitted, eliminating the need for additional division approximations. **(iii) Polynomial Replacement:** In the final stage, each non-polynomial layer is replaced with its polynomial approximation, resulting in a fully polynomial model. Appendix D provides further details on the polynomial approximations used. These approximations are designed to be highly accurate for the HE-friendly weights obtained from the previous stages.

---

**Algorithm 1:** Polynomial Transformer Construction

---

**Input:** A vanilla transformer architecture and hyper-parameters for training
**Output:** A polynomial transformer ready for secure inference
1. **Architectural Modification and Pre-training**: Modify the transformer architecture via
   Eqs. 5 and 4 (stable and Lipschitz HE-friendly variant), and train the new architecture from
   scratch with the same hyper-parameters.
2. **Range-Minimization**: Minimize the input range to GELU, LayerNorm and
   PowerSoftmax layers via the loss function defined in Eq. 8.
3. **Polynomial Replacement**: Replace the inverse function in HE-friendly attention and the
   inverse square root in LayerNorm with polynomial approximations obtained from the
   Goldschmidt method. Replace activations with suitable polynomial approximations (details in
   Appendix D). Incorporate the length-agnostic approximation strategy (Eq. 6).

---

**Reformulate Attention Mask.** Attention masks are a well-known technique used to manipulate self-attention by determining which tokens can attend to each other. Traditional LLMs leverage a mask $M$ for various applications. Notable example is the causal masks, employed for training LLMs via Next-Token Prediction (NTP), a popular self-supervised learning scheme. These standard masking mechanisms are specifically designed for Softmax-based self-attention (masked values were represented by $-\infty$ and used as an as an additive term) and should be reformulated for HE-Friendly Attention, as follows:

$$\text{Masked HE-Friendly Attn}(Q, K, V) = \left( \frac{Q K^T \odot M}{\sqrt{d_k}} \right), \quad M_{i,j} \in [0, 1] \tag{9}$$

**Continual Training.** A significant limitation of Step 1 in Algorithm 1, compared to PTA methods, is the need for retraining, which can be expensive for large transformers trained on extensive datasets. To mitigate this, we propose a complementary procedure to convert standard pre-trained attention layers into PowerSoftmax layers via a short fine-tuning step. Since both attention variants share the same trainable parameters and perform similar (though not identical) computations (as shown in Fig. 1), we initialize the weights of our attention variant from a vanilla pre-trained reference model. Fine-tuning the resulting model reduces the performance gap between the two variants, enabling us to take advantage of the significant computational investment made in these models.

## 5 EXPERIMENTS

We now present an empirical evaluation of our method. Sec. 5.1 introduces our polynomial LLMs and report results on both encrypted and unencrypted data in zero-shot and fine-tuned settings. Sec. 5.2 offers a comprehensive set of ablation studies, providing empirical justifications for the key design decisions of our method, and Sec. 5.3 presents comparisons of our method and others SoTA methods in the domain. Finally, Sec. 5.4 compares the attention matrices generated by the standard Softmax with those produced by our HE-friendly variant, while analyzing the differences between these matrices. The experimental setup is detailed in Appendix B.

### 5.1 POLYNOMIAL LLMs

We experimented with polynomial variants of a causal transformer (GPT) and a bidirectional model.

**Causal Transformer.** For a GPT model, we built upon the Pythia Biderman et al. (2023) family of models, adapting their training procedures, evaluation methodologies, and hyperparameters. Specifically, we trained two models for NTP on the Pile dataset Gao et al. (2020): a small model with 70M parameters and a large model with 1.4B parameters, using continual pretraining (Sec. 4.5).

Table 1: Comparison of zero-shot and 5-shot results between vanilla Transformer and our poly. variant across different model sizes. Original models trained on Pile Gao et al. (2020). Results of non-polynomial models copied from Biderman et al. (2023).

| Dataset | Zero-shot | | | | 5-shot | | | |
| | 1.4B | | 70M | | 1.4B | | 70M | |
| | Orig. | Poly. | Orig. | Poly. | Orig. | Poly. | Orig. | Poly. |
|---|---|---|---|---|---|---|---|---|
| Lambada O. Acc | 0.610 | 0.607 | 0.192 | 0.258 | 0.568 | 0.487 | 0.134 | 0.181 |
| PIQA | 0.720 | 0.710 | 0.598 | 0.592 | 0.725 | 0.720 | 0.582 | 0.597 |
| WinoGrande | 0.566 | 0.562 | 0.492 | 0.503 | 0.570 | 0.568 | 0.499 | 0.505 |
| WSC | 0.442 | 0.395 | 0.365 | 0.365 | 0.365 | 0.548 | 0.365 | 0.452 |
| ARC-Easy | 0.617 | 0.602 | 0.385 | 0.420 | 0.633 | 0.613 | 0.383 | 0.387 |
| ARC-Challenge | 0.272 | 0.265 | 0.162 | 0.185 | 0.276 | 0.277 | 0.178 | 0.183 |
| SciQ | 0.865 | 0.873 | 0.606 | 0.716 | 0.926 | 0.907 | 0.598 | 0.718 |
| LogiQA | 0.221 | 0.217 | 0.235 | 0.210 | 0.230 | 0.222 | 0.250 | 0.238 |

We evaluated these models using the popular lm-evaluation-harness framework. Tab. 1 shows that our models achieve performance comparable to the original models for 5-shot and zero-shot settings. These results mark a significant advancement, as no prior work has introduced polynomial LLMs with demonstrated **ICL or reasoning capabilities**. This is particularly evident on reasoning benchmarks such as the AI2's Reasoning Challenge (ARC), where our models perform competitively.

**Bidirectional Transformer.** For the bidirectional model, we tested our approach on RoBERTa Liu (2019). Starting with a Softmax-based pre-trained transformer, we applied the HE-friendly adaptation using the method described in Sec. 4.5 through continual pre-training on the OpenWebText corpus Gokaslan & Cohen (2019). Then, we fine-tuned our model on 3 datasets from the GLUE benchmark Wang (2018) separately, adapting RoBERTa's fine-tuning process, and

Table 2: Downstream GLUE results for polynomial RoBERTa-Base. Results from Zhang et al. (2024b) are denoted by $\diamond$.

| Model | Dataset | | |
| | SST-2 | QNLI | MNLI |
|---|---|---|---|
| RoBERTa | 94.80 | 92.80 | 87.60 |
| Poly-RoBERTa | 93.35 | 91.62 | 86.93 |
| Nexus (BERT) $\diamond$ | 92.11 | 89.90 | N.A |

finally approximated the non-polynomial components. The results are depicted in Tab. 2, and compared with the work of Zhang et al. (2024a). Full configuration is detailed in App. B.2. These results indicate a degradation of approximately 1% compared to the original RoBERTa performance.

**Latency Over HE.** For benchmarking over encrypted data, we followed the methodology of Zimerman et al. (2024). We first trained a 32-layer polynomial GPT on the Wikitext-103 dataset, then fine-tuned it on a financial news text classification benchmark Muchinguri (2022). The model achieved an accuracy score of 81% over plaintext, reflecting a 10% improvement over the baseline.

Latency profiling for these runs is shown in Fig. 3, measured using HElayers 1.5.4 Aharoni et al. (2023) configured for CKKS with 128-bit security and poly-degree of $2^{16}$. Here, matrix multiplication took $49\% + 18\% = 67\%$ out of which most of it was spent on encoding the plaintext weights. Polynomial approximation accounted for $14\% + 6\% + 4\% = 24\%$ of the total time, where PowerSoftmax took 6% of it. Interestingly, in all polynomial approximations, the most time-consuming HE primitive is the bootstrap operator, confirming that the latency bottleneck is dictated by the polynomials' degree.

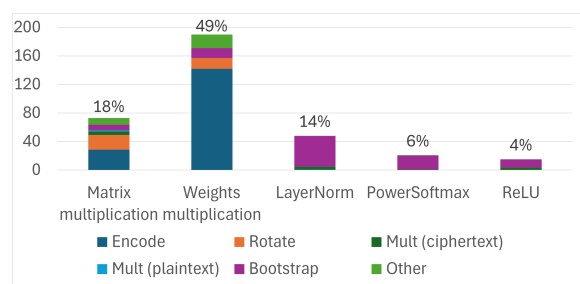

Figure 3: **Latency Over HE:** Time in seconds for main transformer primitives (bars, total = 91%) accumulated across 32 layers. Each bar shows the latency breakdown of the underlying HE operations.

## 5.2 JUSTIFY DESIGN CHOICES

To justify our design choices, we conduct a series of ablations.

**Power-Softmax Attention.** We first compare PowerSoftmax and Softmax outside the context of HE, showing that in addition to being a HE-friendly variant, it also exhibits similar scaling trends as Softmax. Figs. 4 and 4 present comparative visualizations of training curves for various model sizes and datasets (including Pile, Wikitext-103, Text-8, Tiny-Imagenet, CIFAR-100 and CIFAR-10) across both NLP and vision domains, respectively. Although Softmax generally achieves better results, it is evident that by the end of training, most of the gap between the models is reduced, and the scaling laws of the models are relatively similar.

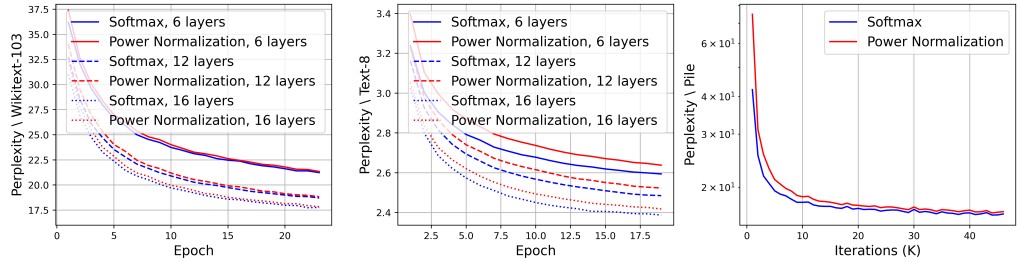

Figure 4: **Training Curves for NTP:** Comparison of test perplexity for transformers with Softmax and power normalization when trained over sevral dataets including Pile, Wikitext-103, and Text-8.

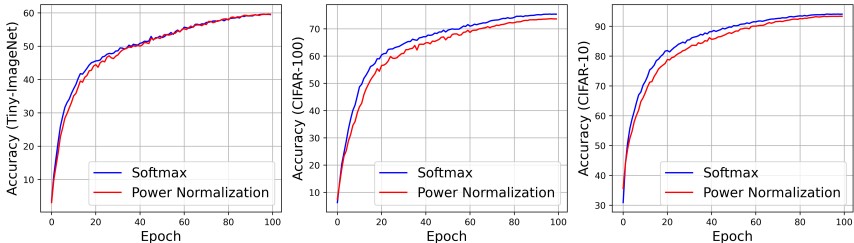

Figure 5: **Results On Vision Tasks.** Training curves for ViT Variants with PowerSoftmax (red) and the Softmax baseline (blue). On the left, results are presented for Tiny-ImageNet and on the middle and right for CIFAR-100 and CIFAR-10 accordingly.

**Stability.** To assess the contribution of our numerically stable variant, we conduct dedicated experiments. In Fig. 6, we provide training curves averaged over 3 seeds for models with 32 layers and hidden dimension size of 1024, trained on 10% of the Wikitext-103 dataset. We compare two Power-Softmax-based transformers with the same training procedure, one with (blue) and one without (red) the stable variant from Eq. 5. As an additional baseline, we trained a vanilla transformer (black). As shown, the stable variant consistently outperforms the Power-Softmax baseline, closing a third of the gap between the Power-Softmax and the softmax baseline. Additionally, we observe that in more challenging regimes, such as training on the full dataset or other datasets, the stable variant is much more robust to optimization issues and less sensitive for hyperpramtaer tunning.

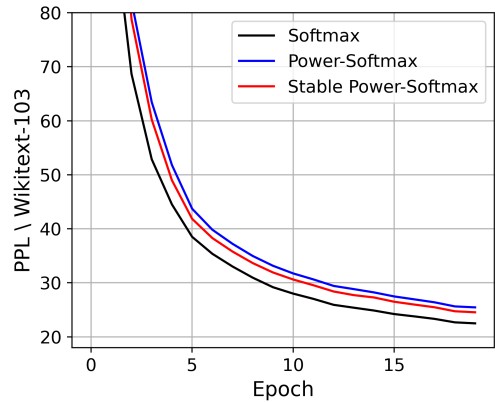

Figure 6: **The Significance of the Stable Variant.** Training curves for NTP on Wikitext for large models .The stable variant (red) consistently outperforms the vanilla PowerSoftmax (blue).

$\epsilon$**-Bounded Division for Softmax.** The HE-friendly attention variant from Eq. 4 proposes adding epsilon to make the approximation problem of division easier, resulting in an approximation of a $\frac{1}{\epsilon^2}$-Lipschitz continuous function. Fig. 7 empirically supports this evidence by showing that the approximation error obtained by the Goldsmith method decreases as epsilon increases. Additionally, Fig. 13 in Appendix C shows that higher values of epsilon improve the training dynamics.

### 5.3 COMPARISONS WITH SoTA METHODS

To the best of our knowledge, only two prior efforts have successfully presented fully polynomial transformers: (i) By Zimerman et al. (2024), which employs the AAT approach, and (ii) Nexus Zhang et al. (2024a), which focuses on the PTA regime. We begin

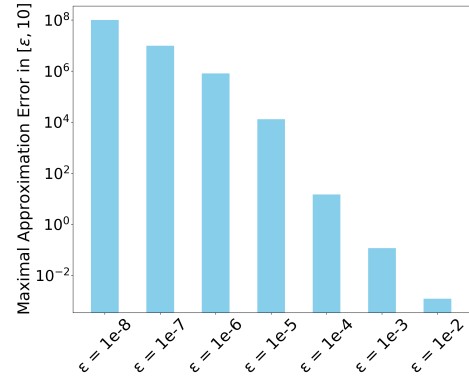

Figure 7: Measuring the polynomial approximation error for different values of $\epsilon$.

by noting that our method exhibits superior scaling properties compared to both these methods. This is evidenced by the fact that both methods concentrated on relatively simple text classification tasks, such as those found in the GLUE benchmark, with or without pre-training. In contrast, our models tackle much more complex tasks, including those that require *reasoning and ICL capabilities*, which are typically associated with LLMs.

Additionally, when operating over encrypted data, our model is significantly more efficient than both of these methods. Specifically, Nexus incorporates three high-degree polynomial approximations at each attention layer, for the exponential, division, and maximum functions, whereas our approach requires only a single non-polynomial division. Regarding (i), we empirically observe that their method exhibits substantially worse scaling properties, particularly for large models, which we were unable to scale up successfully. One possible explanation is that they employ point-wise attention without normalizing attention scores, resulting in a less stable model. We provide a comparison with their method in Fig. 12 in the App. B.2. Moreover, we were unable to train deep transformers with around 32 layers using their method. In terms of the efficiency of secure inference, although both methods include a single non-polynomial operation at each attention head, our method is far more efficient for long contexts. This efficiency gain arises because their method applies an activation function to each element in the attention matrix, resulting in $L^2$ instances of deep polynomials at each attention head. In contrast, our method applies division only once per row, resulting in $L$ deep polynomials which require less HE bootstrap operations.

## 5.4 UNDERSTANDING POWERSOFTMAX THROUGH ATTENTION MATRICES

PowerSoftmax introduces an important hyperparameter $p$ that differentiates it from the traditional Softmax function. To better understand its mechanistic behavior, we examine how the attention matrices evolve with varying values of $p$. Our analysis reveals that as $p$ increases, the resulting attention matrices become more localized as depicted in Fig.9. For instance, by comparing the first column (PowerSoftmax with $p = 4$) with the third column ($p = 12$), we observe a significantly stronger diagonal in the latter, whereas the $p = 4$ model displays a more uniform attention distribution. Additionally, we empirically confirm this pattern by analyzing the average of the mean attention distance (Vig & Belinkov, 2019) per model (i.e., averaged across all the layers and heads) as illustrated in Fig. 8. Moreover, we observe



Figure 8: Measuring the attention mean distance for different transformer variants.

that later layers tend to exhibit more longer-distance relationships compared to earlier layers in both PowerSoftmax and Softmax. This finding is consistent with previous research (Vig & Belinkov, 2019). Additional analysis can be found in the Figures 10 and 11 in the Appendix.

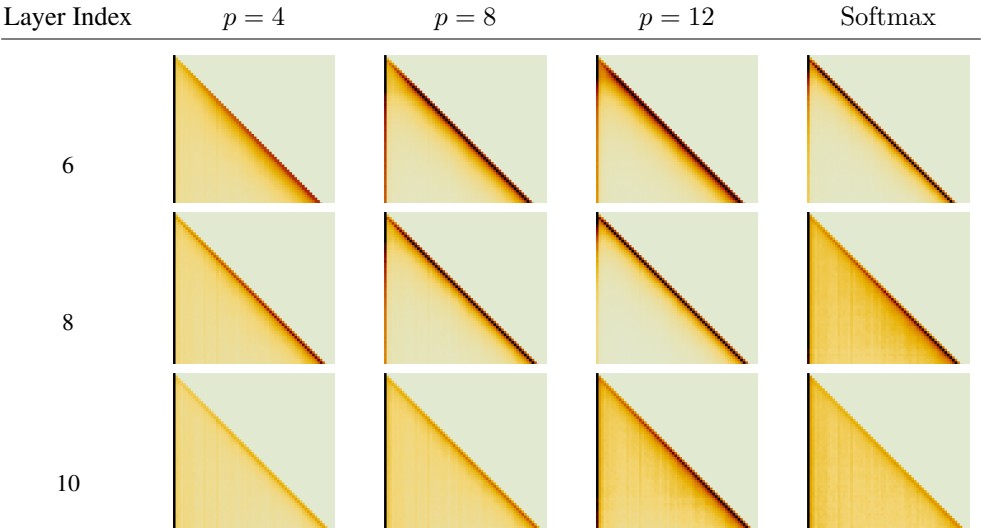

Figure 9: **Visualisation of Averaged Attention Matrices:** Layer Index\Model, where models from left to right are PowerSoftmax with $p = 4, 8, 12$ and Softmax

## 6 CONCLUSION AND LIMITATIONS

We presented a method for training polynomial LLMs with approximately 1.4 billion parameters, significantly larger than those employed in previous works. For that, we introduced a HE-friendly alternative to self-attention, which we demonstrate performs comparably to the original model. This variant allows us to present the first polynomial LLM with zero-shot and reasoning capabilities. Despite the promising results, a full evaluation of the auto-regressive generative abilities of our models in both sequential decoding over plain and encrypted environments has not yet been conducted. For future work, we plan to investigate these aspects further and explore techniques to reduce the model's latency when operating on encrypted data.

## 7 REPRODUCIBILITY STATEMENT

All of our experiments are conducted using the PyTorch framework on public datasets. Furthermore, our codebase is built upon accessible and popular repositories such as the fairseq library for RoBERTa and GPT-NeoX for Pythia models. Additionally, our code for some of the experiments is included as supplementary material. Therefore, we consider our empirical results to be reproducible.

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

## A    ADDITIONAL POLYNOMIAL ATTENTION VISUALIZATION

In Fig. 10 and Fig. 11, we present a visual analysis of attention matrices obtained from both the vanilla Softmax-based models and the corresponding polynomial HE-friendly variants across different layers. Fig. 10 depicts the attention matrices averaged over 3 seeds, all attention heads at a layer, and 1,000 examples. Additionally, to provide a comprehensive view of the attention matrices, Fig. 11 contains random samples of attention matrices. All models rely on a BERT-like 12-layer causal model with a context length of 512, trained on Wikitext-103 for next-token prediction with the same training procedure. We use examples from the test set of Wikitext-103 as input samples.

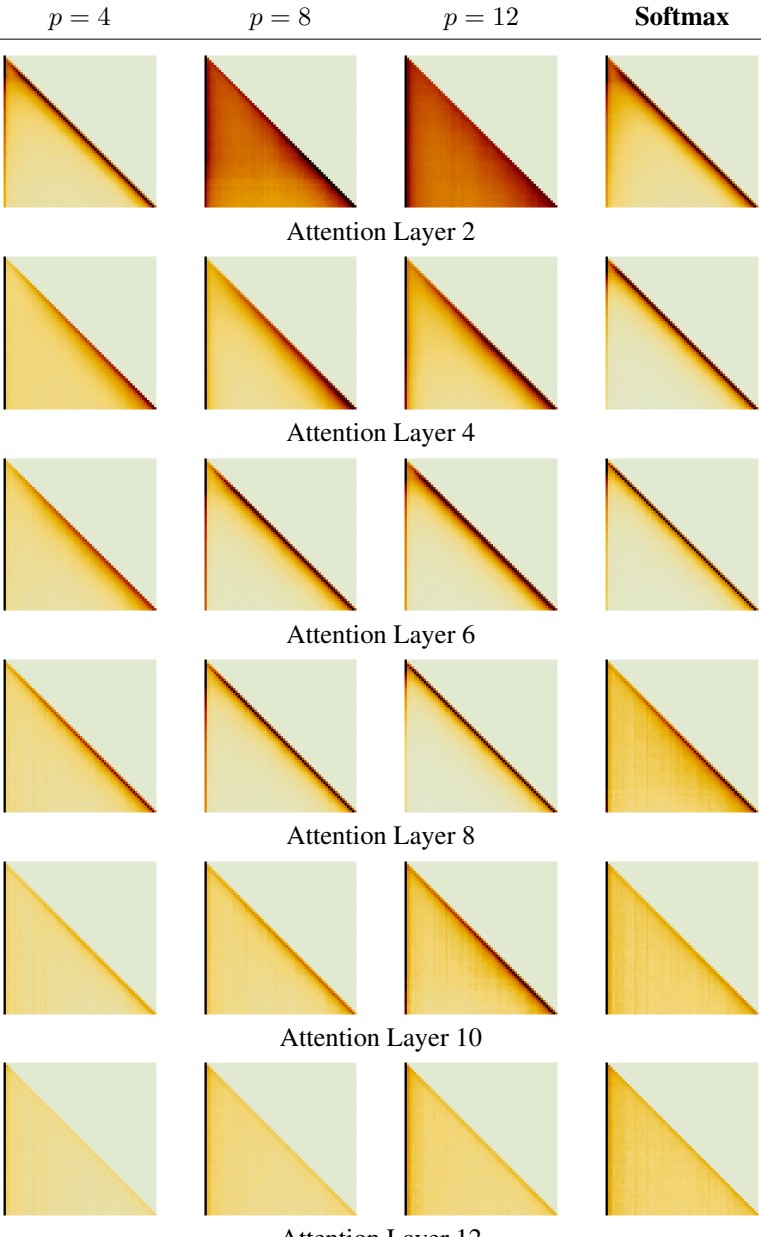

Figure 10: **Visualisation of polynomial average attention matrices:** Models with $P = 4$ (first column) generate more local attention matrices, with reduced mass near the diagonal compared to models with $P = 8$ or $P = 12$, particularly in layers 4-10. In all models, the final layers (rows at the bottom) display more global attention patterns than the middle layers.

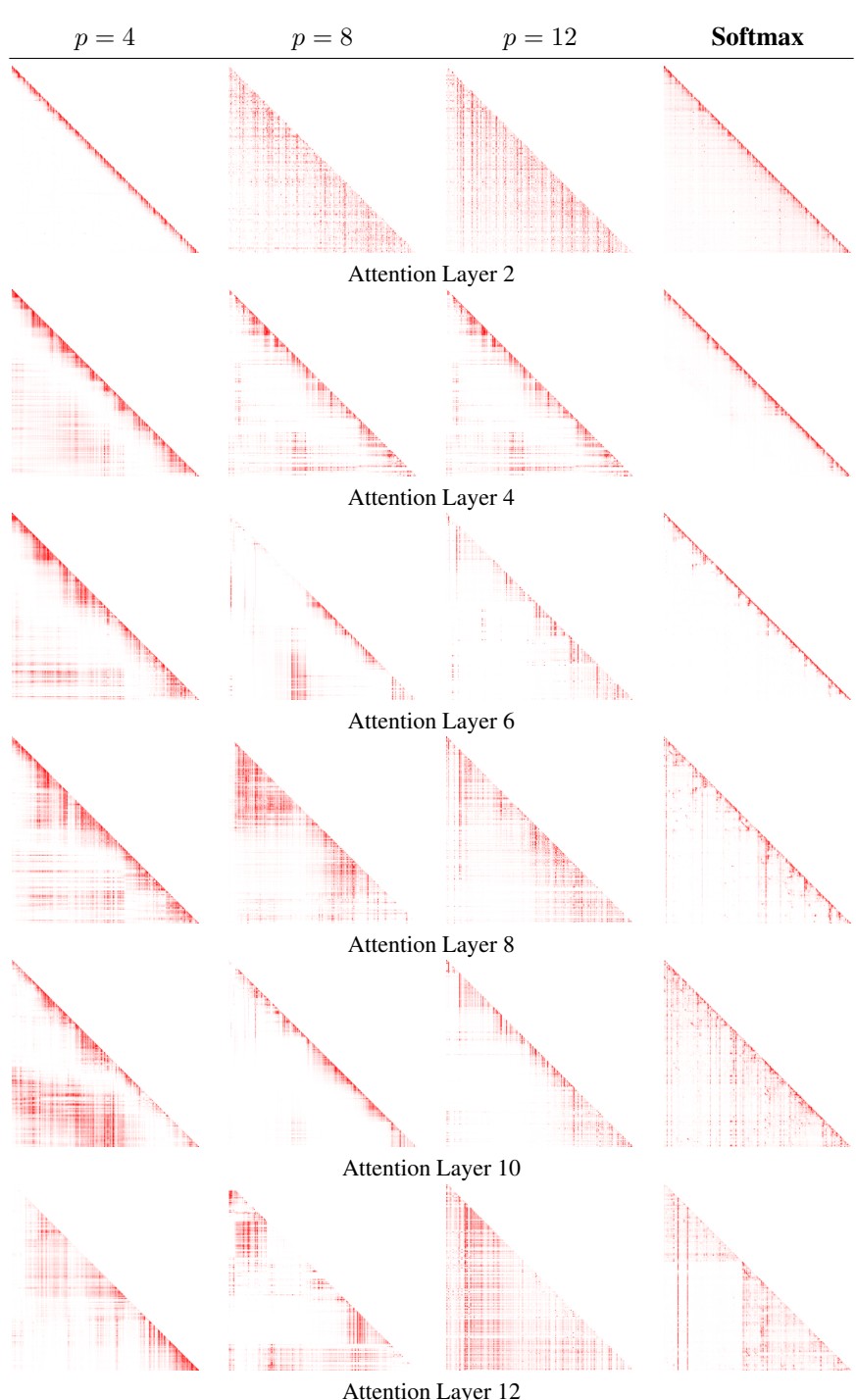

Figure 11: **Visualisation of random samples of polynomial attention matrices:** Although the attention matrices are noisy and a small number of samples may not capture the full distribution trend, the Power-softmax-based models (first three columns) show behavior similar to the original $\mathrm{Softmax}$ (last column). Notably, our attention layers can dynamically adjust focus across different parts of the input, allowing attention heads to freely learn both local and global patterns.

## B EXPERIMENTAL SETUP AND HYPER-PARAMETERS

All training experiments were conducted on public datasets using the PyTorch framework. Results were averaged over three random seeds, with experiments running on two A100 80GB GPUs for a maximum of two days, except for those involving the Pile dataset, which were run for up to three days on eight A100 40GB GPUs.

### B.1 GPT.

We used the framework of neox-gpt[1] Andonian et al. (2023) with its configuration of Pythia to train the 70M and 1.4B models. For this process. The replacement process is done as follows:

1. Load a checkpoint of the pre-trained model.

2. Replace $\mathrm{Softmax}$ with $\mathrm{PowerSoftmax}$, with $p = 4$, and employ continual pre-training of over the Pile dataset for 100 iterations.

3. Finetune the model with range-loss to minimize c and the input to GELU. This process takes around 17K iterations.

4. Apply polynomial approximation.

Table 3 shows the specific hyperparameters used for this process.

| Parameter | GPT 1.4B | GPT 70M | RoBERTa-Base |
|---|---|---|---|
| Sum Power Weights Epsilon | $1e^{-4}$ | 0.001 | $1e^{-4}$ |
| PowerSoftmax Loss Weight ($c$) | $1e^{-4}$ | $1e^{-4}$ | 0.01 |
| GELU Loss Weight | 0.001 | $1e^{-4}$ | 0 |
| Learning Rate | $4e^{-5}$ | $1e^{-4}$ | $1e^{-4}$ |

Table 3: HE-Related Configuration for Pythia 1.4B, 70M, and RoBERTa Models

### B.2 ROBERTA

We employed the RoBERTa framework [2] Ott et al. (2019) and configuration to train and fine-tuned the base model with 125M parameters for three GLUE tasks: SST-2, QNLI, and MNLI. The process was carried out as follows:

1. Load a checkpoint of the pre-trained base model.

2. Replace $\mathrm{Softmax}$ with $\mathrm{PowerSoftmax}$, with $p = 6$, and continual pre-training the model on the OpenWebText dataset for 1250 iterations.

3. Fine-tune the model individually for each of the three GLUE tasks for up to 10 epochs. This fine-tuning followed the procedure described in the original RoBERTa paper, except for substituting the Tanh activation function in the classification head with a Sigmoid, which we found to perform better under HE.

4. Perform an additional fine-tuning step using range-loss with $\mathrm{PowerSoftmax}$ loss weight for 10 epochs. The GELU ranges were narrow enough and did not require tuning.

5. Apply polynomial approximation.

We reported accuracy results in table 2. See Table 3 for the specific hyperparamerters.

Additionally, we train RoBERTa models with 12 layers from scratch over the Wikitext-103 benchmark for three types of attention: (i) Softmax (black), (ii) our Power-Softmax (blue), and (iii) the Scaled-ReLU (red) attention baseline of Zimerman et al. (2024), all using the same training procedure and hyperparameters optimized for the vanilla Softmax-based Transformer. Training Curves

---

[1] https://github.com/EleutherAI/gpt-neox
[2] https://github.com/facebookresearch/fairseq/blob/main/examples/roberta

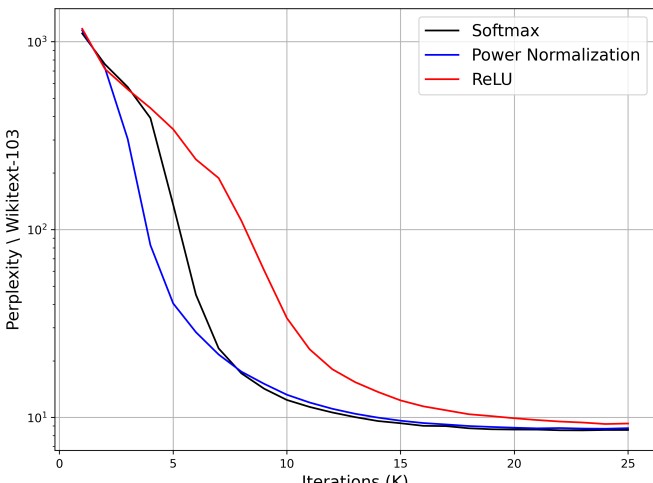

Figure 12: Comparison of training curves for 12-layer RoBERTa models with different attention mechanisms on the Wikitext-103 benchmark. The Power-Softmax variant (blue) converges faster than Softmax (black), while the Scaled-ReLU baseline (red) underperforms. Curves are averaged over three seeds.

are averaged over three seeds and presented in Fig. 12. As shown, the Scaled ReLU variant is not competitive with the variants that employ proportional normalization. While $\mathrm{Softmax}$ achieves better final results, it converges slightly slower than the $\mathrm{PowerSoftmax}$ variant. With the implementation of early stopping, the models achieved average perplexity of 8.48 for $\mathrm{Softmax}$, 8.69 for $\mathrm{PowerSoftmax}$, and the Scaled ReLU lag behind with 9.12.

## C   ADDITIONAL ABLATION STUDIES

To gain a clearer understanding of the impact of $\epsilon$ in $\mathrm{PowerSoftmax}$-based attention models, we trained several models using different values of $\epsilon$. As shown in Figure 13, our variants demonstrate robustness across various $\epsilon$ values in terms of training dynamics. However, Figure 7 shows that for larger values of $\epsilon$, the resulting approximation function for division becomes easier, and we consider these settings (as an example $\epsilon = 1e - 2$) to be preferred.

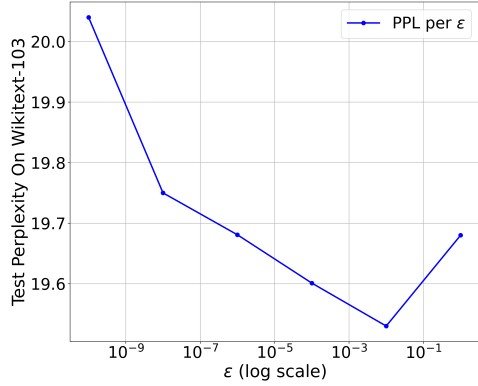

Figure 13: The impact of different values of $\epsilon$ on training dynamics of $\mathrm{PowerSoftmax}$-based models

## D  OUR POLYNOMIAL APPROXIMATIONS

Our PowerSoftmax-based transformers utilize three polynomial approximations. For the division in PowerSoftmax and the $\frac{1}{\sqrt{x}}$ function in LayerNorm, we apply the Goldschmidt approximation, following previous work in the domain Zimerman et al. (2024); Zhang et al. (2024a). For the GELU approximation, we use the following identity to reduce the problem to approximating the Sigmoid function, which has been extensively explored in previous research in the HE domain.

$$GELU(x) = x \cdot \text{Sigmoid}(1.702 \cdot x) \tag{10}$$

