# OpenReview forum: "PowerSoftmax: Towards secure LLM Inference Over Encrypted Data"
_ICLR.cc/2025/Conference — Submitted to ICLR 2025_

### Official Review · Reviewer_Hr7u · 2024-10-29

**Soundness:** 3
**Presentation:** 2
**Contribution:** 2
**Rating:** 3
**Confidence:** 3

**Summary:**

This paper proposes a new HE-friendly self-attention layer for privacy-preserving transformer inference. The authors introduce two new variants of HE-friendly attention for model training and inference, respectively, and proposed a polynomial transformer construction algorithm. Finally, the authors conduct empirical studies to validate the effectiveness of the proposed method.

**Strengths:**

The main advantages can be listed as follows:

1.	The paper proposes a new HE-friendly attention by analysing the important properties the attention layer should have. The proposed variants of attention have better numerical stability for training, and can be transformed into polynomial.
2.	The paper conducts experiments and ablation studies on 8 datasets to show the effectiveness of the proposed method.

**Weaknesses:**

Despite the strengths, there are some issues with the paper as follows:

1.	The writing could be further improved. For example, $\epsilon’$ in Eqn. (5) and $\odot$ in Eqn. (9) are not defined ($\odot$ in Line 77 only defined for two ciphertext numbers, rather than matrices). Additionally, "x" in Line 258 should be written as "$x$". Moreover, Furthermore, Line 219 references Eqn. (6) without any prior description, making it challenging to follow the content.
2.	I am concerned about the novelty of this paper’s contributions. The main contribution appears to be the proposed attention layer, as described in Eqns. (4) and (6). However, other techniques such as the supplementary training procedure and polynomial approximations, are adapted from previous works rather than newly introduced.
3.	The paper’s significance may be limited without a detailed latency comparison for HE-based inference. Although the authors claim to present the first private LLM with 1B parameters, the primary challenge for HE-based large-scale models lies in the high latency of HE operations (in other words, building large-scale models is trivial if latency is disregarded). The paper lacks inference accuracy and latency comparisons with prior studies. Including such comparisons would significantly enhance the evaluation of the proposed method's effectiveness.

**Questions:**

1.	What is the inference latency of the proposed method, and how does it compare with previous HE-based neural network methods?
2.	In line 218, it is stated that “it is clear that Eq. (2) and Eq. (6) can lead to training instability…”. Could the authors provide a more detailed explanation regarding this point?

---

> ### Author Response · Authors · 2024-11-26
> **Response to Hr7u**
>
> Thank you for taking the time to provide such informative feedback.
>
> Weaknesses:
>
> > W.1. The writing could be further improved. For example, ϵ′ in Eqn. (5) and ⊙ in Eqn. (9) are not defined (⊙ in Line 77 only defined for two ciphertext numbers, rather than matrices). Additionally, "x" in Line 258 should be written as "x". Moreover, Furthermore, Line 219 references Eqn. (6) without any prior description, making it challenging to follow the content.
>
> We sincerely thank the reviewer for pointing out these issues and providing detailed feedback to improve the clarity of our paper. We have addressed the concerns in the revised version.
>
> > W.2. I am concerned about the novelty of this paper’s contributions..
>
> We respectfully disagree with the reviewer that the novelty is limited. Please see our response in the shared response and let us know if you have further concerns.
>
> > W.3. The paper’s significance may be limited without a detailed latency comparison for HE-based inference. Although the authors claim to present the first private LLM with 1B parameters, the primary challenge for HE-based large-scale models lies in the high latency of HE operations (in other words, building large-scale models is trivial if latency is disregarded). The paper lacks inference accuracy and latency comparisons with prior studies. Including such comparisons would significantly enhance the evaluation of the proposed method's effectiveness.
>
> Answer: Please see our global comment regarding the concerns about HE. Additionally, it may be the case that using a large enough approximation, any model could become a polynomial model. However, the practicality of this claim has never been tested or proven. On the contrary, training polynomial models is considered a difficult task (see references in the text). We demonstrate for the first time that it is possible, while also suggesting ways to improve the practicality of polynomial models under FHE.
>
> ___
> Questions:
>
> > Q.1. What is the inference latency of the proposed method, and how does it compare with previous HE-based neural network methods?
>
> Answer: Please see our global comment about HE concern.
>
> > Q.2. In line 218, it is stated that “it is clear that Eq. (2) and Eq. (6) can lead to training instability…”. Could the authors provide a more detailed explanation regarding this point?
>
> Similar to Softmax, numerical issues can arise because exponential functions or polynomial functions like x^p amplify large inputs (greater than 1) while shrinking small inputs (less than 1). This creates a wide dynamic range where large values dominate, and small values contribute negligibly, leading to a potential loss of significance and resulting in numerical instability. Similar to the log-sum-exp trick, our stable variant mitigates this by dividing by the maximum absolute value, ensuring the values remain within a numerically stable range.

---

> > ### Comment · Reviewer_Hr7u · 2024-11-30
> >
> > Thank you for the detailed response. However, I still have concerns about the contributions of the work. It appears to be a combination of the new softmax function and several other tricks, and the new loss function in Eqn. (8) seems to be an $l_\infty$-norm regularizer.  Additionally, the writing could be improved. Hence, I will maintain my current score.

---

### Official Review · Reviewer_ZaGC · 2024-11-03

**Soundness:** 3
**Presentation:** 2
**Contribution:** 2
**Rating:** 3
**Confidence:** 4

**Summary:**

This paper introduces a variant of the self-attention module that reduces the overhead of Homomorphic Encryption (HE) computation during private Transformer inference. The authors first replace the exponential scaling in the original SoftMax function with polynomial scaling in the proposed PowerSoftMax function. The authors further introduce a stable variant of PowerSoftMax with Lipschitz division and division invariant, and a length-agnostic variant of PowerSoftMax with pre-computation. Experiments show that the proposed PowerSoftMax-based Transformers maintain reasoning and in-context learning (ICL) capabilities. The authors also offer a latency breakdown.

**Strengths:**

1. The research question is important and timely. Protecting the privacy of user data is important in LLM inference.
2. The proposed methods effectively improve the efficiency of the self-attention module during HE-based private LLM inference.
3. The evaluation and analysis are detailed. The authors present the source code and detailed results.

**Weaknesses:**

1. The novelty is limited. Replacing non-polynomial functions (e.g., the exponential function) with polynomial counterparts is a commonly used technique in private LLM inference works. While the authors also propose to integrate techniques like Lipschitz division and division invariant (Section 4.3 and 4.4) to enhance the stability of the proposed PowerSoftMax function, strategies like pre-computation (Section 4.5) to improve the efficiency of the division operation, many of these techniques seem incremental upon existing works.
2. The scalability is questionable. While the authors show that the proposed PowerSoftMax-based Transformer can achieve similar reasoning capability as the SoftMax-based counterparts, the proposed method requires training from scratch. Retraining the LLMs is often impractical. Despite the authors trying to show the proposed methods are effective for a 32-layer Transformer, I am still concerned about how the proposed method can scale to large models.
3. The threat model is not clear. Although the authors briefly described the problem setting in Section 3, the security threats remain unclear to me. For example, it remains unclear why the model weights should be encrypted if there are only two parties. The threat model should be clarified with respect to the capability of the involved parties (e.g., client, model provider, and server) and the capability of the adversaries. Additionally, the security description about HE in untrusted environments is not accurate (line 109). In untrusted environments, HE also suffers from verifiability issues and side-channel attacks.
4. The HE experiments are not clear or comprehensive. While the main objective of this paper is HE-based private LLM inference, the only experiment related to HE I can find is in Figure 3, which is not clear to me. There is also a lack of comparisons with related works or baselines, making the evaluation of the proposed method not comprehensive.
5. There is some inconsistency in the presentation. For example, in Section 4.3, the authors introduce the division invariant property of the proposed PowerSoftmax. Yet, in Figure 2, it shows subtraction invariant ($x-\max(|x|)$). The readability should be enhanced.

**Questions:**

1. The breakdown in Figure 3 does not include the latency of the GELU function. Is the GELU function not used? Is SoftMax the main bottleneck during private inference?
2. If the model weights need to be encoded and encrypted, is this a one-time process that can be performed offline?
3. What is the multiplicative depth set in the HE experiments? Is bootstrapping performed after each operation (since every bar in Figure 3 contains bootstrapping)?
4. How do you decide the parameter $p$, the degree of the polynomial, in PowerSoftMax in practice? The choice of $p$ is pivotal since it determines the efficiency-utility trade-offs.

---

> ### Author Response · Authors · 2024-11-26
> **Response to ZaGC**
>
> We thank the reviewer for their comments.
>
> Weaknesses:
>
> > W.1. The novelty is limited.
>
> We respectfully disagree with the reviewer that the novelty is limited. Please see our response in the shared response and let us know if you have further concerns.
>
>
> > W.2. The scalability is questionable… The proposed method requires training from scratch. Retraining the LLMs is often impractical.
>
> We would like to clarify an important point: our approach does not require training LLMs from scratch. Instead, we employ **continual pretraining** (lines 325-332),  where we start from a pretrained model, replace Softmax with our proposed PowerSoftmax-based attention, and fine-tune the model with the same hyperparameters as of the original model until convergence, as described in Algorithm 1. Since PowerSoftmax preserves the mechanism of the original Softmax, this replacement can be seamlessly integrated after pretraining, allowing us to retain the core functionality and knowledge encoded in the pretrained model.
>
> This is in contrast to paper [1], which replaces Softmax with an elementwise attention mechanism, specifically ReLU. Such a replacement alters the fundamental behavior of the attention mechanism, requiring training from scratch and affecting model scalability.
>
> By leveraging continual pretraining, our approach significantly reduces computational costs while ensuring scalability. As a result, our method is capable of effectively handling larger models, including those with more than 32 layers.
>
> Moreover, continual pretraining facilitates incremental updates to existing models, making the integration of our method both feasible and practical for real-world applications.
>
> [1] Zimerman et al. "Converting transformers to polynomial form for secure inference over homomorphic encryption", ICML 2024.
>
>
> > W.3. The threat model is not clear.
>
> Answer. Please see our global answer for the security concern. We added the threat model description in Appendix H.
>
> > W.4. The HE experiments are not clear or comprehensive.
>
> Answer: Figure 3 reports the contribution of every component to the latency when computing under FHE. Specifically, it shows that PowerSoftmax consumes only 6% out of the total run. Please see our global comment about HE concern regarding a comparison with other Softmax approximation techniques and the reason that we choose to present only this experiment. Note that until this paper, there were no benchmarks to compare with except for Nexus, where we explained in the paper why our PowerSoftmax approach is better.
>
> > W.5. There is some inconsistency in the presentation.
>
> We thank the reviewer for catching this typo, which we have corrected in the revised manuscript. To improve clarity, we have also performed additional professional proofreading. The revised manuscript was uploaded.
>
> ___
>
> Questions:
>
> > The breakdown in Figure 3 does not include the latency of the GELU function. Is the GELU function not used? Is SoftMax the main bottleneck during private inference?
>
> This evaluation was performed with ReLU activations instead of GeLU activations. However, this choice is not critical, and the experiment can be easily reproduced with other activation functions if the reviewer considers it important.
>
> As the number of tokens in a prompt increases, the size of the attention matrix grows quadratically, leading to a significantly higher computational cost for the softmax function compared to other operations that scale linearly. Consequently, improving the attention mechanism is a critical objective for polynomial transformers in secure inference, especially for efficiently handling long context.
>
>
> > If the model weights need to be encoded and encrypted, is this a one-time process that can be performed offline?
>
> Answer: Yes and No—there is an interesting tradeoff here. Due to the number of parameters and the choice of FHE packing, the size of the encoded and encrypted model can be quite large, ranging from MB to TB. When the model size exceeds the capacity of the CPU/GPU cache, it is more efficient to encode and encrypt the model on the fly. The decision of whether to use a preprocessed input or to evaluate it dynamically is an interesting and orthogonal research topic to ours, and it should be addressed by the developers of FHE compilers like HElayers.

---

> > ### Author Response · Authors · 2024-11-26
> > **Cont.**
> >
> > > What is the multiplicative depth set in the HE experiments? Is bootstrapping performed after each operation (since every bar in Figure 3 contains bootstrapping)?
> >
> > Answer: We configured HElayers to use the HEaaN library for CKKS, with its FGb parameter set specified as follows: log(QP) = 1555, N = 2^16, L = 9, h = 192, \lambda = 128. As indicated, the number of levels between bootstraps is L = 9. Additional details can be found in the HEaaN documentation. Note that the HElayers SDK has its own bootstrap policy, which may decide not to use all 9 levels before performing bootstrapping. This decision is made automatically and is abstracted from users of HElayers to simplify the development of FHE applications.
> >
> >
> > > How do you decide the parameter p, the degree of the polynomial, in PowerSoftMax in practice?
> >
> > Our empirical analysis shows that P = 4 and P = 8 yield the best training dynamics. To minimize multiplication depth while maintaining accuracy, we select P = 4 for its optimal balance of efficiency and performance during training. We have included a detailed description of this strategy, along with an analysis of the impact of P, in a new appendix (see Appendix E) in the revised manuscript.

---

> ### Comment · Reviewer_ZaGC · 2024-11-29
>
> Thanks to the authors for their efforts and detailed response. I have the following comments.
>
> > The nolvety.
>
> I acknowledge the authors' analysis on the properties of Softmax-based attention. However, the key properties of Softmax-based attention have been widely explored in related works such as [1] (see their second motivation).
>
> > The scalability.
>
> I appreciate the authors explanations. The fine-tuning strategy makes sense to me. However, if the training-from-scratch (in Algorithm 1) is not actually used in this paper, it would be more clear the authors replace training-from-scratch (e.g., those in line 307) with fine-tuning. Additionally, how many epochs and time are needed to fine-tune the PowerSoftmax? These overheads should be taken into consideration when comparing with NEXUS, since NEXUS is a PTA method which does not need fine-tuning.
>
> > Specifically, in this paper, the reviewers can consider only the simple case where the model weights are generated by the server and thus are unencrypted (while the query is still encrypted by the data-owner).
>
> Even if the proposed methods use the common threat models in prior-arts (e.g., those in the client-server and client-server-model provider settings), I do not think the threat model should be omitted. The current statement in Section 3 remains ambiguous. At least more clarification should be given in the Appendix.
>
> Also, the authors suggest the reviewers to focus on the client-server setting where the weights are unencrypted. If this setting is the main focus of the paper, all experiments should be based on this setting. However, encoding the weights becomes the main bottleneck in Figure 3, which is confusing to me.
>
> The model weights encryption in Q2 can also be clarified along with the threat model.
>
> > Presentation.
>
> In addition to the unclear threat model, there are many other points I find unclear and confusing with respect to the presentation. Figure 2 (middle) shows subtraction invariance, which appears inconsistent with Equation 5. Additionally, in line 440, 'with' should be red instead of blue to align with Figure 6. I believe the manuscript can benefit from a careful proofreading and revision.
>
>
> > The GELU function
>
> The authors explicitly mention GELU in line 309 in Algorithm 1 and in the Appendix D. It should be clarified for which experiments GELU are used and for which ReLU are used. Replacing GELU with ReLU without clarification is misleading. The only reference about ReLU I can find is the scaled-ReLU attention baseline.
>
> > HE experiments.
>
> I understand the authors have used HElayers for the experiments. However, it remains unclear why not directly experiment on HE libraries like HEaaN. As mentioned by the authors, HElayers 'may decide not to use all 9 levels before performing bootstrapping'. This actually keeps the readers from learning the benefits from PowerSoftMax, i.e., the improved depth consumption and reduced number of bootstrapping.
>
> Additionally, directly comparing the runtime with NEXUS and [2] is not convincing. NEXUS is based on the SEAL library and [2] is based on the HEaaN library. Also, NEXUS and [2] do not need fine-tuning, while fine-tuning is indispensable for PowerSoftMax. More importantly, utility/efficiency trade-offs should be demonstrated instead of comparing the runtime only. I acknowledge the comparing with these works is not straightforward, and I do not mean the authors must show these comparison results now.
>
> I do not see Appendix H and E in your revision. Please check if the submission has been correctly updated. I will maintain my rating at this point.
>
> [1] Zeng et al., "Mpcvit: Searching for accurate and efficient mpc-friendly vision transformer with heterogeneous attention." ICCV 2023.
>
> [2] Cho et al., "Fast and Accurate Homomorphic Softmax Evaluation." CCS 2024.

---

> > ### Author Response · Authors · 2024-12-01
> >
> > Thank you @ZaGC for reading our rebuttal and for the response.
> >
> >
> > **From your response, we understand that a mistake was made and that our updated PDF, which includes Appendices E-H and the proofreading we performed, did not make its way to the OpenReview website.** Since the editing period has passed, we can only confirm that all the requested changes (both from the review and below) have been incorporated into our paper.
> >
> >
> > 1. Indeed, some properties of Softmax have also been explored in other papers (clearly, we are not the first to research Softmax). Still,
> >
> >
> >       a) [1] did not explore our variants of Softmax, which are optimized for FHE.
> >
> >       b) [1] only considered toy datasets, such as CIFAR-10 and Tiny-ImageNet. Note that [3] (which is cited in our paper) also            considered 'substituting Softmax with a scaled ReLU.' However, as we wrote:
> >
> >
> >  > They demonstrated a 100M-parameter polynomial transformer pretrained on WikiText-103 for secure classification tasks using HE.
> >
> >
> > > This is evidenced by the fact that both methods concentrated on relatively simple text classification tasks, such as those found in the GLUE benchmark, with or without pre-training.
> >
> >
> > > In contrast, our models tackle much more complex tasks, including those that require reasoning and ICL capabilities, which are typically associated with LLMs.
> >
> >
> > Summary: Our novel PowerSoftmax variants achieve better results compared to prior art.
> >
> >
> > 2. Threat model: We understand the reviewer's view about the threat model and as mentioned it is already included in an appendix.
> >
> >
> > 3. Our experiments are based on the client-server settings with unencrypted weights.
> >
> >
> > 4. Indeed, encoding the weights for models with a large number of parameters is currently a bottleneck, which we hope will be addressed in the future. While the encoding operation is generally fast, the number of times it is called is significantly larger compared to operations like Softmax (e.g., L = 32 times over one ciphertext). **We emphasize again that our work does not aim to optimize all primitives of encrypted transformers, but rather focuses only on Softmax, as demonstrated in both our paper and the rebuttal comments**.
> >
> >
> > 5. Our revision also includes our answer to Q2.
> >
> >
> > 6. Presentation/GELU-ReLU comment. As mentioned, our fixed and proofread version was unfortunately not updated on the website. However, all of your suggestions, as well as additional changes, have been incorporated into the revision. Regarding the GELU-ReLU comparison, we had two training settings. The first setting, 'training from scratch' on the Wikitext-103 dataset, used the ReLU activation. The second setting, for larger models (Pythia 1.4B), was trained on The Pile dataset, where we applied a fine-tuning approach. Since Pythia uses GELU, we followed their settings.
> >
> >
> > 7. Regarding:
> >
> > > I understand the authors have used HElayers for the experiments. However, it remains unclear why not directly experiment on HE libraries like HEaaN.
> >
> > We are puzzled by the reviewer’s question. Does the reviewer believe we should avoid using PyTorch and implement all of our experiments directly in NumPy? Or, even better, implement them directly in C++?
> >
> > HELayers uses libraries like HEaaN as its underlying FHE engine but hides the complexity of writing complex FHE circuits. It allows data scientists (who are not cryptographers) to simply provide an ONNX file as input and that's it. If we had used HEaaN, we would have had to implement every transformer ourselves (similar to writing in NumPy), which could introduce implementation biases regarding whether we used the HEaaN library correctly. To avoid such bias, we used an off-the-shelf SDK (such as PyTorch) that ensures full reproducibility of our results. Therefore, it is unclear to us why the reviewer asks us to avoid using state-of-the-art solutions.
> >
> >
> > > This actually keeps the readers from learning the benefits from PowerSoftmax, i.e., the improved depth consumption and reduced number of bootstrapping.
> >
> >
> > We respectfully disagree. PowerSoftmax only involves one division operation, so its multiplication depth is well-defined. For example, let’s say its multiplication depth is 27 (raising the input to the 4th power requires 2 multiplications, and using 22 iterations of the Goldschmidt algorithm, together with 3 additional scalar multiplications, results in a total of 27 multiplications). In comparison, [2] requires at least a multiplication depth of 35. Whether someone uses bootstrapping after every multiplication i.e., 27 times (resp., 35 times) or after every 9 multiplications, i.e., 3 times (resp. 4 times) is an implementation decision, and is orthogonal to our research, in any case our solution improves performance.
> >
> >
> > [3] Itamar Zimerman, Moran Baruch, Nir Drucker, Gilad Ezov, Omri Soceanu, and Lior Wolf. Converting transformers to polynomial form for secure inference over homomorphic encryption. ICML 2024, https://proceedings.mlr.press/v235/zimerman24a.html.

---

> > > ### Author Response · Authors · 2024-12-01
> > >
> > > 8. Regarding the scaling concern and the comparison to NEXUS, when using a fine-tuning approach instead of training from scratch, the number of iterations required to obtain a fully-trained model is less than 10% of the full training process. While approaches like NEXUS and similar works eliminate training time entirely, we believe that optimizing inference latency is more critical. This is because training is a one-time process, whereas inference occurs repeatedly many times during the model’s usage.

---

> ### Comment · Reviewer_ZaGC · 2024-12-02
> **Thank you for your clarification**
>
> Thanks to the authors for their efforts and clarification. HElayers is a powerful tool that makes HE more accessible to more researchers and is widely employed in the community. I do not mean that the authors must avoid using HElayers or any other off-the-shelf SDKs. I believe one of the main benefits of PowerSoftmax is the improved depth consumption and reduced number of bootstrapping, which should be highlighted.
>
> Following the example provided by the authors, one can use bootstrapping after every 12 multiplications. Consider bootstrapping restores the full multiplicative budget [2], 3 bootstrapping operations are needed for both PowerSoftmax and [2]. The number of multiplications are reduced from 35 in [2] to 27 in PowerSoftmax. When using HE libraries like HEaaN, the reduced number of multiplications clearly contributes to reduced depth consumption. Therefore, with such circuit-level optimization, it is possible to perform 9 additional multiplications before the next bootstrapping.
>
> But when using HElayers, it is not clear to me whether the additional multiplications can be performed before the next bootstrapping, since HElayers 'may decide not to use all levels before performing bootstrapping'. That is, the actual depth consumption and number of bootstrapping are determined by the circuit-level optimization (e.g., PowerSoftmax) and the compiler-level optimization (e.g., HElayers).
>
> It might be possible that the HElayers-based implementation can still effectively demonstrate the circuit-level optimizations in PowerSoftmax. But I believe more clarification and discussion are needed for this potential impact brought by the implementation choice, especially for readers not familiar with HElayers.

---

### Official Review · Reviewer_SRrQ · 2024-11-05

**Soundness:** 2
**Presentation:** 3
**Contribution:** 2
**Rating:** 3
**Confidence:** 4

**Summary:**

The most challenging aspect of implementing AI models based on homomorphic encryption is the implementation of nonlinear functions, such as softmax, in an encrypted state. This study aims to modify the Transformer model by reducing nonlinearity in the softmax function using a power function, thereby decreasing the homomorphic encryption computation load without compromising performance.

**Strengths:**

1. A very wide range of experiments was conducted.
- They experimented with various models, including BERT-based models, GPT-based models, and ViT-based models, and included many different tasks.
- The experiments also included models with different parameter sizes (e.g., 70M, 135M, 1.4B) and deep models with 32 layers.
- Since softmax is the main focus, they provided a comprehensive comparison of epsilon values used in softmax as well as a comparison with standard softmax, achieving thorough experimental results.
- They averaged results using three seeds, and all parameters used were recorded, indicating excellent reproducibility.

**Weaknesses:**

1. HE-based implementation was not conducted.
- They only referenced the time it takes for softmax in other papers, stating that their work could reduce that time by 6%.

2. Contribution seems limited.
- They introduced three modifications to the revised power-softmax:

       * Lipschitz Division

       * Stable PowerSoftmax

       * Length-Agnostic Range

- The first is simply adding epsilon to prevent division by zero, which, though not explicitly stated, is generally used in all implementations.
- The second scales the numerator and denominator of softmax, which is conceptually the same as dividing by maxX in regular softmax.
- The third, which involves using different functions for training and inference, was innovative. While producing the same output, it effectively narrows the range by applying the mean-based function for inference.
- Other methods like layernorm and range minimization are existing methods. In summary, the contributions boil down to introducing power-softmax and separating inference and training with length-agnostic range.

3. There is a reliability concern with the results of the zero-shot and 5-shot experiments.
- The paper states the following regarding the similar results in zero-shot and 5-shot experiments:

      "These results mark a significant advancement, as no prior work has introduced polynomial LLMs with demonstrated ICL or reasoning capabilities. This is particularly evident on reasoning benchmarks such as AI2’s Reasoning Challenge (ARC), where our models perform competitively."

- However, the zero-shot and 5-shot results alone are insufficient to accurately gauge the effectiveness of the model. Due to the simplification of complex non-linear functions into a linear form, the initial convergence may be faster due to computational advantages, but it may not fully retain the non-linearity and expressive power of the original softmax. Showing fully-trained results would be more appropriate.

**Questions:**

See the weakness.

---

> ### Author Response · Authors · 2024-11-26
> **Response to SRrQ**
>
> Thank you for your effort and for taking the time to help us improve our paper.
>
> Weaknesses:
>
> > W.1. HE-based implementation was not conducted.
>
> Answer: We partially disagree, Line 382 reports our HE-based experiments. More data is provided as an answer in the global comment to the HE concern.
>
> > W.2. Contribution seems limited.
>
> Answer: We respectfully disagree with the reviewer that the contribution is limited. Please see our response in the shared response and let us know if you have further concerns.
>
> > W.3. There is a reliability concern with the results of the zero-shot and 5-shot experiments.
>
> First, the main results of our work, as presented in Tables 1 and 2, are based on fully-trained models. These results closely align with the performance of the original Pythia models of the same size and are directly compared against the benchmarks reported in their paper (see Table 8 and 12).
>
> Additionally, we agree that to demonstrate the comprehensive effectiveness of the PowerSoftmax variant, it is important to (i) provide results of fully-trained models and (ii) present training curves, as is standard practice in this domain. Our paper already addresses these points by including detailed evaluations across more than 10 datasets, multiple modalities, several training regimes (full training, fine-tuning, zero-shot), and various model sizes, as shown in Tables 1 and 2 and Figures 4 and 5.
>
> Finally, to further address the reviewer’s concerns and enhance clarity, we have created a new table (Table 4, Appendix F) in the revised manuscript. This table summarizes the final performance of the fully trained models, derived from Figures 4 and 5, across six distinct datasets and several model sizes.
>
> For all of these reasons, we kindly request that you reconsider this point.

---

### Official Review · Reviewer_aaxZ · 2024-11-09

**Soundness:** 3
**Presentation:** 3
**Contribution:** 4
**Rating:** 6
**Confidence:** 4

**Summary:**

This paper introduces a modified version of the Transformer architecture, adapting it to the constraints of Homomorphic Encryption (HE) through a power-based self-attention variant that is compatible with polynomial representations. The proposed model achieves performance comparable to Softmax-based Transformers across multiple benchmarks while preserving the core design characteristics of self-attention. Additionally, the paper presents variants that incorporate length-agnostic approximations and enhanced numerical stability. This approach provides a more HE-friendly Transformer solution than previous methods, enabling efficient scalability to large language models with 32 layers and 1.4 billion parameters.

**Strengths:**

This paper proposes an HE-friendly self-attention variant, which minimizes non-polynomial operations while preserving the core principles of the attention mechanism. The approach is further extended by introducing a stable numerical training method and a length-agnostic computation strategy for inference, supporting secure and scalable inference. Leveraging this technique, the authors develop a polynomial variant of RoBERTa and train the largest polynomial model to date, comprising 32 Transformer layers and approximately 1.4 billion parameters.
Notably, this work expands the Approximation-Aware Training (AAT) approach for Transformers by replacing Softmax with a polynomial-friendly alternative, closely replicating its functionality. The proposed method enhances model performance and scalability while remaining compatible with homomorphic encryption operations.

**Weaknesses:**

The authors should provide critical information on latency and resource utilization, which is essential for assessing the feasibility of applying homomorphic encryption (HE) to transformers. Specifically, there is insufficient detail regarding the HE settings and parameters used, making it unclear what type of simulation was conducted. Additionally, there is no information on the security model under which the HE simulations were performed. The description of the homomorphic implementation of matrix operations, including ciphertext-ciphertext multiplication beyond Softmax, needs to be included.

**Questions:**

The authors claim that their method is more efficient than existing approaches, even though no information is provided on latency. On what basis is this claimed efficiency over previous methods established? Additionally, what homomorphic encryption parameters were used in the simulations?

**Details Of Ethics Concerns:**

No concern.

---

> ### Author Response · Authors · 2024-11-26
> **Response to aaxZ**
>
> We are grateful for your detailed review, which has guided us in improving our paper.
>
> Weaknesses:
>
> > W.1. The authors should provide critical information on latency and resource utilization, which is essential for assessing the feasibility of applying homomorphic encryption (HE) to transformers. Specifically, there is insufficient detail regarding the HE settings and parameters used, making it unclear what type of simulation was conducted. Additionally, there is no information on the security model under which the HE simulations were performed. The description of the homomorphic implementation of matrix operations, including ciphertext-ciphertext multiplication beyond Softmax, needs to be included.
>
> Answer: Please see our global comment about HE concern.
> ___
> Questions:
>
>
> > Q.1. The authors claim that their method is more efficient than existing approaches, even though no information is provided on latency. On what basis is this claimed efficiency over previous methods established?
>
> Answer: From the text: "When operating over encrypted data, our model is significantly more efficient than both of these methods. Specifically, Nexus incorporates three high-degree polynomial approximations at each attention layer for the exponential, division, and maximum functions, whereas our approach requires only a single non-polynomial division." Please see a comparison of PowerSoftmax in the global comments.
>
> > Q.2. Additionally, what homomorphic encryption parameters were used in the simulations?
>
> Answer: Please see our global comment about HE concern.

---

### Author Response · Authors · 2024-11-26
**An answer to the novelty concerns**

**Novelty and Contribution:** Several reviewers have expressed concerns about the marginal novelty of our work. Hence, we would like to clarify our position by specifying the novelties in our method:

While one might argue that our approach simply replaces exponents with powers in the Softmax function or substitutes non-polynomial operations with polynomial counterparts, this perspective significantly oversimplifies our contributions. We propose to reframe the discussion as follows:

First, **we identify and distil the key properties of Softmax-based attention** (normalization of attention scores, super-linear scaling, monotonicity, and order preservation). Leveraging these principles, we propose a novel mechanism specifically tailored for FHE environments. This mechanism adopts distinct forms to address two primary challenges: ensuring stable training over plaintext data at scale and enabling efficient secure inference over encrypted data.

Polynomials, while necessary for FHE compatibility, are inherently sensitive and often struggle when trained from scratch. To overcome this limitation, our method **retains the non-polynomial division during training** to ensure that attention scores remain precisely normalized (bounded, summing to 1 to prevent vanishing). This design choice, **coupled with our stable variant, is critical for maintaining the robustness and scalability required for SOTA transformers at scale**, as demonstrated in Figure 6.

For inference, we simplify the approximation by replacing the exponential function with a carefully selected polynomial, x^p, which is more efficient in both space and computational complexity that standard approximations. To achieve this, we decouple the exponential scaling and proportional normalization components of the Softmax function. While this substitution does not perfectly preserve all Softmax properties, it effectively captures their trends, offering a practical trade-off, as discussed in Section 4.1.

This polynomial activation applies to **every element in the attention matrix, which contains a quadratic (in sequence length) number of elements. Therefore, this optimization is crucial for efficiently handling long sequences** and avoiding computational bottlenecks during activation computation over the attention matrix.

Novel Extensions:

(i) The stable variant is both novel and critical for FHE. Naive computation requires non-polynomial operations, such as maximum and division, which are infeasible over FHE. Since approximating them with high-degree polynomials is inefficient, **we address this challenge with a unique loss function (Eq. 8) during fine-tuning. It aligns the stable and original attention variants for inference while eliminating the need for non-polynomial normalization** (lines 288–297). This approach goes beyond a simple adaptation of the log-sum-exp trick.

(ii) The length-agnostic variant addresses the challenge of adjusting polynomials for varying sequence lengths. **Since polynomial approximation depends heavily on the domain range, each sequence length requires a distinct approximation. By leveraging the non-encrypted nature of sequence length during inference over FHE, we compute its inverse directly, ensuring efficiency and consistency across different lengths.**

(iii) Finally, the Lipschitz Division variant prevents division by zero and, **more importantly, simplifies the approximation problem, making it feasible with low-rank polynomials** (Figure 7).

For all these reasons, we kindly ask the reviewers to reevaluate this point.

---

> ### Author Response · Authors · 2024-11-26
> **An answer to the HE concern**
>
> Our paper is the first to report polynomial LLMs with 1.4B parameters for tasks relevant to LLMs, as depicted in Table 1, addressing a key obstacle to making FHE-encrypted LLM solutions feasible. We acknowledge that challenges remain in making FHE practical, particularly due to performance and memory gaps between inference over encrypted data and plaintext. However, Rome wasn’t built in a day. While FHE has shown approximately 2x annual speedup through software improvements alone, it will take time before it becomes practical for complex tasks such as LLMs. In our view, this should not impede AI research. On the contrary, we hope to close the FHE gap from above, while FHE software and hardware accelerations will narrow the gap from below.
>
> Focusing on AI advancements, our paper introduces a new primitive called PowerSoftmax, and our HE measurements support the need to optimize this primitive. To avoid burdening readers, we opted for simplicity by using an off-the-shelf library (HElayers) and running our model as-is. Therefore, our paper does not delve into complex or unrelated design decisions, such as packing strategies or matrix multiplication algorithms. First, these decisions are abstracted away by the HElayers compiler, and second, this approach allows the reader to focus on the core AI contribution. In a similar way, we do not report the algorithms used by PyTorch, by GCC, or Clang, we trust the compilers. Further, note that our approach is orthogonal to the implementation of models over FHE and will improve the latency of any implementation simply by replacing Softmax with PowerSoftmax in the compiler. This is why we prefer to compare only the Softmax latency, rather than the overall latency, as the latter could mislead the readers.
>
> With that said, we will add to the paper that the latency reported by HElayers on one A100 80GB GPU for our PowerSoftmax over 32 layers of 128x128 took only 16 seconds. In contrast, prior SOTA -- NEXUS reported that Softmax using 4 (instead of 1) A100 GPUs over 128x128 blocks for 12 (instead of 32) layers took 1.15 seconds * 32 (batch size) = 36.8 seconds, x2.3 slower. Moreover, following the submission of this paper, another study addressing Softmax was published [a]. This paper introduces a new Softmax approximation. Compared to [a], our method reduces multiplication depth by at least 23% when replacing Softmax with our PowerSoftmax in fine-tuned models, where the latency reported by [a] on an NVIDIA RTX-6000 for 128x128 * 32(layers) + 1x32768 inputs was 90 seconds. again, much slower.
>
> As stated in the paper, all our experiments used HElayers 1.5.4 (Aharoni et al. (2023)), configured for CKKS with 128-bit security and polynomial degree of 2^16. We also note that the underlying CKKS library was HEaaN, using its FGb parameter set with the following specific values: log(QP) = 1555, N = 2^16, L = 9, h = 192, \lambda = 128. Additional details can be found in the HEaaN documentation.
>
> Finally, it is important to emphasize that our paper reports actual measurements over FHE-encrypted data, rather than simulations, establishing the feasibility of our solution under FHE.
>
>
> [a] Wonhee Cho, Guillaume Hanrot, Taeseong Kim, Minje Park, and Damien Stehlé. 2024. Fast and Accurate Homomorphic Softmax Evaluation. arXiv preprint arXiv:2410.11184 (2024). https://arxiv.org/abs/2410.11184

---

> > ### Author Response · Authors · 2024-11-26
> > **An answer to the security concerns**
> >
> > Section 3, lists two threat-model examples for secure inference for LLMs over HE: 1) using encrypted weights; or 2) using encrypted input samples (queries). When considering a 2-party scenario with FHE, it is commonly assumed that one party is the (untrusted) server who holds the model (encrypted or not) and the other party is the data-owner who performs the query and would like to avoid revealing the query to the server. The decision whether the server holds the model encrypted or not depends on whether a third-party the model-owner agree to share the data with the server.
> >
> > Our proposed solution is orthogonal to the above decision, which eventually only affects latency. As there are many prior-art that dealt with this threat model, for brevity, we have decided to omit it, we will add a reference to [a] for further clarification. Specifically, in this paper, the reviewers can consider only the simple case where the model weights are generated by the server and thus are unencrypted (while the query is still encrypted by the data-owner).
> >
> > In FHE-based applications it is common to assume that the server is semi-honest, i.e., that it follows the protocol (integrity of the computation is assumed) but it cannot learn anything about the query content of the data-owner. We agree with @ZaGC that the word “untrusted” can be conceived as too broad and we will clarify in the text that by untrusted server we refer only to semi-honest servers. With that saying, please note that no side-channel issues exist with encrypted data sent to the server, because by definition the data is encrypted using an IND-CPA scheme and any adversary against such a scheme cannot extract information about the data with advantage more than negligible.
> >
> > @aaxZ, to add on the above, Line 387 states that we used HElayers configured for CKKS with 128-bit security and poly-degree of 2^16. This is a commonly used configuration for DNN inference over FHE.
> >
> > [a] Allon Adir, Ehud Aharoni, Nir Drucker, Ronen Levy, Hayim Shaul, Omri Soceanu, Homomorphic Encryption for Data Science (HE4DS) [Chapter 3], https://doi.org/10.1007/978-3-031-65494-7

---

### Meta-Review · Area_Chair_nNhu · 2024-12-19

**Metareview:**

The reviewers were split about this paper and did not come to a consensus: on one hand they appreciated the motivation of the work and the (non-HE) experiments conducted, on the other they had issues with (a) paper clarity, and (b) limited novelty. Only two reviewers responded to the author feedback (ZaGC, with a detailed discussion & Hr7u, briefly to maintain score). No reviewers engaged in further discussion of the paper. After going through the paper and the discussion I have decided to vote to reject based on the above issues. Specifically, for (a) nearly all reviewers expressed confusion about the HE experiments, prompting the authors to write a long global response about this. Even though the authors did clarify these things in the rebuttal, I found that, without these details, the paper has not been modified sufficiently to resolve these confusions. For (b), the reviewers argued that simply replacing exponents with powers in the softmax and adding polynomial approximation is not enough of a contribution to accepted at ICLR. The authors responded with another long global response. This response makes their contribution clearer, but requires a significant rewrite of the paper to modify the story in this way. Given all of the above, I believe this work should be rejected at this time. Once these things and other issues mentioned in the reviews are addressed in an updated version, the work will be much improved.

**Additional Comments On Reviewer Discussion:**

Please see the above meta-review for details on this.

---

### Decision · Program_Chairs · 2025-01-22

Reject